# Prenatal Constant Light Exposure Induces Behavioral Deficits in Male and Female Rat Offspring: Effects of Prenatal Melatonin Treatment

**DOI:** 10.3390/ijms26031036

**Published:** 2025-01-25

**Authors:** Tsveta Stoyanova, Hristina Nocheva, Zlatina Nenchovska, Desislava Krushovlieva, Petya Ivanova, Jana Tchekalarova

**Affiliations:** 1Institute of Neurobiology, Bulgarian Academy of Sciences, 1113 Sofia, Bulgaria; ts.stoyanova@inb.bas.bg (T.S.); z.nenchovska@inb.bas.bg (Z.N.); daisyveko@gmail.com (D.K.); ivanova.petya91@gmail.com (P.I.); 2Department of Physiology and Pathophysiology, Faculty of Medicine, Medical University of Sofia, 1431 Sofia, Bulgaria; dr_inna@yahoo.com

**Keywords:** prenatal constant light exposure, offspring, sexual dimorphism, anxiety and depression, melatonin, rat

## Abstract

Prenatal constant light exposure (CLE) impaired the anxiety response and circadian rhythms of testicular enzymes in adult male rat offspring, while melatonin corrected these deficiencies. However, the mechanism by which CLE induces these long-term behavioral consequences and the impact of melatonin system have not been examined. The aim of the present study was to investigate the effects of prenatal CLE and melatonin treatment on anxiety- and depression-like behaviors, and the melatonin system in male and female adult rat offspring. Six groups of male and female rat offspring (P60) exposed to either light/dark (LD) or CL regimes, and treated with vehicle or melatonin (10 mg/kg, s.c.) were evaluated for anxiety by open field (OF), elevated plus maze (EPM), and light/dark (LD) tests, and depressive-like response by splash test and sucrose preference test. Plasma adrenocorticotropic hormone (ACTH), corticosterone (CORT) and melatonin expression, and hippocampal MT1A and MT1b receptor expression were assessed by ELISA. Prenatal CLE induced behavioral deficits and elevated plasma CORT levels, while melatonin levels, their circadian rhythmicity, and hippocampal MT receptor expression were not altered in male and female offspring in the CLE regime. However, prenatal melatonin treatment corrected behavioral deficits in a sex-specific manner by up-regulating hippocampal MT receptors, even without altering systemic melatonin levels or normalizing CORT in either sex. The results of this study suggest critical insights into how prenatal environmental factors and therapeutic interventions shape physiological and behavioral outcomes.

## 1. Introduction

Numerous experimental and clinical studies have shown that intense psychological and emotional experiences during pregnancy can have varying degrees of impact on the normal course of labor and delivery [1], as well as on the psychosomatic development of the offspring [2,3]. A major focus concerns stress processes in pregnancy and the effects on preterm birth and low birth weight. The current evidence points to pregnancy anxiety as a key risk factor in the etiology of preterm birth, and chronic stress and depression in the etiology of low birth weight. The ability of mammals to cope with physical and psychoemotional stress is due to the activation of a cascade of neuroendocrine responses called the hypothalamic–pituitary–adrenal (HPA) axis. Potential stress-inducing factors activate the HPA and culminate in the adrenal gland synthesizing and secreting glucocorticosteroids (GCSs). GCSs affect virtually all tissues by facilitating the body’s stress response. Acute increases in GCS levels under stress are associated with physiological and behavioral changes that are beneficial and essential to the experience [4]. However, sustained high levels of GCS in chronic stressful situations or disease states are detrimental and increase the risk of stress-related pathology [4,5,6]. Elevated levels of the stress hormone cortisol during pregnancy are thought to correlate with low birth weight and cardiovascular, metabolic, neuroendocrine, and behavioral disorders in adulthood [7]. Stress during pregnancy disrupts the circadian rhythms of the sleep–wake cycle, nutrition, and hormone secretion, which increases the risk of developing various diseases in the offspring [8].

Intrauterine development is a critical period during which the organism is susceptible to stress-related pathologies that may manifest later in life. If the mother is stressed during pregnancy, the offspring are more likely to develop emotional problems, including an increased risk of attention deficit hyperactivity disorder, symptoms of depression, and anxiety and reduced cognitive function [9]. These data suggest that various prenatal exposures can negatively affect intrauterine development and have lasting adverse effects on offspring [10,11,12].

The pineal gland is the primary source of circulating plasma melatonin, and its circadian secretion modulates the circadian dynamics of several physiological functions [13]. Melatonin is an indoleamine with multiple functions. The oscillations of its secretion in mammals, controlled by an endogenous circadian timing system, regulate the body’s light/dark cycle, thereby organizing its diurnal and seasonal rhythmicity [14]. Endogenous biological rhythms can be altered by environmental factors known as synchronizers. One of the most important synchronizers is light, whose control over rhythms is exerted by a neural pathway through the retinohypothalamic tract and by a humoral pathway through melatonin secretion. Melatonin deficiency caused by constant light exposure (CLE) leads to a disruption of the circadian rhythm of several physiological and biochemical parameters, including disruption of the synchronizing role of the hormone on endocrine and metabolic functions in the body, and dysfunction of the myocardium, blood vessels, and kidneys [15,16]. Melatonin is thought to play an essential role in the regulation of fetal circadian rhythms as it is one of the few hormones whose molecule crosses the placental barrier unchanged [17,18]. During intrauterine development, the pineal gland of the fetus does not synthesize melatonin and the required amount is obtained from the maternal organism [19]. Hormone binding sites have been identified in the suprachiasmatic nucleus (SCN) of fetuses from day 18 of gestation [20]. The maternal program regulates the development of the circadian system during the fetal and neonatal period [18,21]. Data on melatonin deficiency due to CLE in pregnant mothers and the subsequent physiological and biochemical changes in their offspring are very scarce. It has been found that prolonged LE during the prenatal period leads to changes in the diurnal oxidations of malondialdehyde (MDA) and lactate dehydrogenase in the testes of male rat offspring. This effect was accompanied by an anxiogenic effect and changes in copulatory behavior. The administration of melatonin during pregnancy corrected these abnormalities in the offspring [18].

In the present study, we investigated the consequences of prenatal CLE and the effect of melatonin treatment on the behavioral responses and hormonal status of adult male and female rat offspring. We found that, while CLE during rat pregnancy caused behavioral deficits accompanied by elevated plasma corticosterone in both adult male and female offspring, the melatonin system was intact. However, prenatal melatonin treatment ameliorated increased anxiety- and depressive-like responses via up-regulated MT receptors in the hippocampus.

## 2. Results

### 2.1. Melatonin Treatment Corrected Anxiety in Both Offspring Sexes with Prenatal CLE

Prenatal CLE reduced the time spent in the central zone vs. total time compared to being exposed to the light/dark (LD) regime in rats treated with vehicle (veh) (*p* = 0.026 (male) and *p* = 0.05 (female), respectively) in the OF test (Figure 1). Treatment with melatonin alleviated the anxiety-like response of male and female offspring with a history of prenatal CLE (*p* = 0.016, male light–light–melatonin (LL-mel) vs. male LL-veh; *p* = 0.037, female LL-mel vs. female LL-veh group, respectively).

Anxiety-like responses in male and female offspring exposed to prenatal CLE were also examined in the elevated plus maze (EPM) test. The time spent in the aversive zone (open arms) was significantly reduced in both the male LL-veh group vs. the LD-veh group (*p* = 0.0015) and the female LL-veh rats vs. the LD-veh group (*p* = 0.0022) (Figure 2). Prenatal melatonin treatment prolonged the time spent in the aversive open arm (open arm vs. total time) in male offspring (*p* < 0.001, vs. LL-veh group) but not in female offspring (*p* = 0.177, vs. LL-veh group).

An increased anxiety-like response was detected in both male and female offspring with a history of CLE compared to the matched LD controls in the light/dark test (LDT). Thus, male and female LL-veh groups had (1) increased latency to stay in the dark compartment (*p* < 0.001, vs. male and female LD-veh groups, respectively) (Figure 3A,B); (2) reduced number crossing into the light compartment (*p* < 0.001, vs. male LD-veh group and *p* = 0.013, vs. female LD-veh group) (Figure 3C,D); (3) shorter time spent in the light compartment (*p* < 0.001, vs. male LD-veh) (Figure 3E) and (*p* = 0.0028, vs. female LD-veh) (Figure 3F). Melatonin treatment of pregnant mothers subjected throughout the gestational period to a CLE regime corrected the increased anxiety response to control level both in male (*p* = 0.0094, *p* = 0.0075, *p* < 0.001, vs. LL-veh) (A,C,E) and female (*p* < 0.001, *p* = 0.0264 vs. LL-veh) (B,D) offspring.

A reduced preference for sucrose solution was observed in LL-veh males (*p* = 0.0025, vs. LD-veh group) and females (*p* < 0.001, vs. LD-veh group), suggesting a depressive-like behavior. Prenatal injection of melatonin prevented anhedonia in male and female offspring (*p* = 0.021 and *p* < 0.001, respectively) compared to the LL-veh model group injected with veh alone (Figure 4).

The depressive nature of the behavior was also confirmed by the splash test. Reduced grooming time was reported in both male and female LL-veh offspring (*p* < 0.001, male and female LL-veh vs. LD-veh group) (Figure 5A,B). Grooming time was statistically increased in the male and female groups treated with melatonin (*p* < 0.001 and *p* = 0.0045, male and female LL-mel vs. LL-veh group), confirming the beneficial effect of melatonin on depressive-type behavior in the offspring of rats from pregnant mothers exposed to 24 h of light.

### 2.2. Prenatal Melatonin Treatment Caused Up-Regulation of Melatonin MT1A and MT1b Receptors in the Hippocampus, but Did Not Affect Prenatal CLE-Induced Elevated Plasma Corticosterone Levels

While the plasma adrenocorticotropic hormone (ACTH) was not changed as a result of prenatal CLE in both male and female offspring (*p* > 0.05) (Figure 6A,B), the corticosterone (CORT) levels were significantly elevated in the two offspring sexes compared to the LD-veh matched group (*p* = 0.0234, male LL-veh and *p* < 0.001, female LL-veh) (Figure 6C,D). Pretreatment with melatonin throughout pregnancy did not suppress the prenatal CLE model-induced increase in plasma levels of this hormone in either sex (*p* < 0.001, male and female LL-mel vs. matched LL-veh group).

Prenatal CLE did not alter the expression of the hormone melatonin in the plasma of male and female rats (Figure 7A,B). Diurnal oscillation in the melatonin expression was shown in both the male LD-veh and LL-veh groups (*p* = 0.0064 and *p* < 0.001, vs. light phase) and the female LD-veh and LL-veh groups (*p* = 0.0111 and *p* < 0.001, vs. light phase), respectively.

While the prenatal CLE did not affect the expression of the two melatonin MT_1A_ and MT_1b_ receptors in the hippocampus of male and female offspring (*p* > 0.05, vs. LD-veh matched group), melatonin treatment up-regulated both MT_1A_ and MT_1b_ receptors in the hippocampus (*p* = 0.029, vs. male LD-veh group, and *p* = 0.0018, vs. male LD-veh group, respectively) (Figure 8A,C). The melatonin-induced up-regulation of MT_2b_ receptors was even stronger in female offspring with a history of prenatal CLE (*p* = 0.02, vs. matched LD-veh group and *p* = 0.0089, LL-veh group) (Figure 8D).

## 3. Discussion

Prenatal stress can have significant effects on fetal development, affecting the brain and behavior in ways that often manifest as sexual dimorphism [22]. Emotional dysregulation increases the risk of psychiatric disorders, including anxiety, depression, and mood disorders, which often occurs in offspring. One of the main factors leading to negative consequences for embryo development is the presence of disturbances in the mother’s emotional state, which could lead to severe psychosomatic disorders in postnatal development such as depression, post-traumatic stress disorder, and anxiety in the offspring [1].

In the present paper, we report that CLE throughout the gestational period of the pregnant rat results in long-term adverse behavioral and biochemical consequences in both adult male and female offspring. However, no sex dimorphism in anxiety- and depressive-like behavior was observed, although sexually dimorphic effects of prenatal stress could explain sex differences in the prevalence and expression of certain psychiatric disorders [23]. In addition, other stress-induced manipulations during the pre- or postnatal period have shown that females may be more vulnerable to mood and anxiety disorders [22,24]. Prenatal stress may disrupt the organization of the HPA axis and affect sex hormone production later in life. This disruption may influence emotional regulation differently in males and females due to their non-identical hormonal profiles. Females may have a phase-dependent difference in the long-term effects of prenatal stress, whereas males may lack such hormonal buffering. This may partially explain the sex dimorphism in the prevalence of stress-related psychiatric disorders, with females being more susceptible to conditions such as depression and anxiety due to the modulation of sex hormones. In addition, we showed that, while ACTH levels were not altered, plasma CORT levels were elevated in both adult male and female offspring compared to controls, which may partially explain the lack of sex dimorphism in emotional responses in them.

Melatonin is one of the few hormones that crosses the placental barrier unchanged, suggesting its essential role in regulating the fetal circadian rhythms [17,18]. During intrauterine development, the fetal pineal gland does not synthesize melatonin, and the required amount is obtained from the mother’s body. Melatonin treatment of pregnant mothers has a positive and corrective effect on the mentioned psychosomatic disorders in the offspring, both male and female. There is still not much literature and research available regarding the effect of prenatal CLE and the role of the melatonin system in male and female offspring. We found that, although adult offspring with a history of prenatal CLE showed an impaired emotional status and elevated CORT levels, plasma melatonin levels were not different from controls. The limitation of this study is that melatonin levels in plasma were not examined in the pregnant rats under CLE. Our data suggest that the model of prenatal CLE in pregnant rats is characterized by a melatonin deficiency as previously reported by our team in adult male rats under the CLE regime [25]. However, the impaired mechanism of melatonin release from the pineal gland was not inherited in the offspring, suggesting intact circadian rhythms of physiological and biochemical functions. Exposure to CL during the prenatal period was found to result in changes in daily oxidations of MDA and lactate dehydrogenase in the testes of male rat offspring, and was accompanied by anxiogenic effects and changes in copulatory behavior [18]. Melatonin supplementation during pregnancy corrected these abnormalities in the offspring. In the present study, we demonstrated that prenatal melatonin treatment restored the impaired emotional status in adult offspring. This beneficial effect on anxiety was stronger in male offspring, whereas no anxious response was observed in EPM and time spent in the light compartment in the LDT in female offspring.

Moreover, while melatonin MT_1b_ receptor expression was up-regulated in both male and female offspring with prenatal CLE, MT_1A_ receptors were increased only in male offspring with melatonin treatment. This sex difference in melatonin receptors may be related to the sexual dimorphism in the effect of melatonin on altered anxiety levels. Among its receptor subtypes, MT_1b_ receptors have been studied for their potential association with anxiety and stress regulation [26]. MT_1b_ receptors are found in the central nervous system, including brain regions involved in anxiety and stress-related changes, such as the amygdala, hippocampus, and prefrontal cortex. Studies in animal models with mutations or knockouts of MT_1b_ receptors have shown altered anxiety-like behaviors, often with increased response to stress. However, in C3H/HeN mice with genetic deletion of both MT_1A_ and/or MT_1b_ receptors, behavioral deficits associated with impaired anxiety levels and depressive-like responses were observed [27]. This finding suggests that dysregulation of both melatonin receptors MT_1A_/MT_2b_ may contribute to the underlying mechanisms of emotional disturbance (anxiety and depression). Therefore, the weaker effect of melatonin supplementation in female offspring with prenatal CLE could be explained by the lack of effect of prenatal melatonin treatment on the expression of MT_1A_ receptors.

## 4. Materials and Methods

### 4.1. The Animals and Experimental Design

Male and female 3-month-old Wistar rats were housed under normal conditions (temperature 21 ± 1 °C; 50–60% humidity; 3–4 per cage). After a 1-week acclimation period, female rats were impregnated and separated into individual cages after placement of a vaginal plug to indicate pregnancy. Pregnant dams were divided into the following groups according to light regime—light/dark (LD) and light/light (LL)—and treatment method—vehicle (veh) and melatonin (mel): (1) control pregnant LD-veh group (male n = 8 and female n = 9), under artificial LD regime (12-h light/dark cycle, 08:00 a.m. on) and treated with veh; (2) LL-veh group exposed to constant light (120 lux), provided by a 40 W white fluorescent tube, and treated with veh (male n = 8 and female n = 8); (3) LL-mel group, under LL regime and treated with mel (male n = 6 and female n = 6). Rats were injected subcutaneously (s.c.) with mel at a dose of 10 mg/kg from G0 to G21, two hours before the dark phase, based on available literature data and our previous studies [25]. Matched groups (LD-veh and LL-veh) were treated in the same manner with vehicle. After postnatal day 21, same-sex offspring were divided into groups according to lighting regime and prenatal treatment. Each control and the experimental group consisted of at least six rats containing pups from 4 or 5 litters. The experiments were conducted according to the guidelines of the Council of the European Communities of 24 November 1986 (86/609/EEC). The project was approved by the Bulgarian Food Safety Authority No. 338/19.11.2022.

### 4.2. Behavioral Tests

#### 4.2.1. Open Field Test (OF)

The test rat was placed in the center of a gray polystyrene box with the dimensions (100 × 100 cm × 60 cm). The trajectory of movement and visit in the central aversive zone was recorded for 5 min using software (SMART PanLab software 3.0, Harvard Apparatus, Holliston, MA, USA).

#### 4.2.2. Elevated Plus Maze Test (EPM)

This test was performed on the EPM apparatus, which consisted of two open (50 × 10 cm) and two closed (50 × 10 × 50 cm) arms. The rat was placed in the central zone facing one of the two open arms. The time spent in the open arms was recorded by a video tracking system (SMART PanLab software, Harvard Apparatus, USA) for 5 min.

#### 4.2.3. Light/Dark Test (LDT)

The test was performed in a box divided by a partition into two compartments—dark and light. At the beginning of the test, the rat was placed in the light zone. The following parameters were evaluated for 5 min: (1) latency to enter the light zone; (2) total time spent in the light zone, which reflects anxiety-like behavior; (3) number of crossings from the light to the dark compartment.

#### 4.2.4. Sucrose Preference Test (SPT)

The SPT was used to assess the anhedonia-like response in rats. During the test, each test rat was placed in an individual cage and acclimated to drinking from two identical plastic graduated bottles of tap water (100 mL) for seven days. For 48 h, a pretest was performed in which the water in one of the two bottles was replaced with a 1% sucrose solution. During the 24 h test, the amount of water tested relative to the sucrose solution was recorded. Preference for the sucrose solution was calculated as the percentage of sucrose consumed out of the total liquid consumed during the light and dark phases.

#### 4.2.5. Splash Test

The splash test was used to assess depressive-like behavior in rats. The test was performed by spraying a 10% sucrose solution on the dorsal surface of the rodent’s body, and grooming behavior was measured in terms by licking the fur to remove the solution. Decreased grooming behavior indicated depressive symptoms and was measured in seconds for 5 min.

### 4.3. ELISA Test

#### 4.3.1. Measurement of Plasma Adrenocorticotropic Hormone (ACTH) and Corticosterone (CORT) Levels

The blood taken from the tail vein was carefully collected into Vacutainer^®^ Tubes with EDTA, and centrifuged at 6000 rpm for 15 min at 4 °C. The plasma was isolated and stored at −70 °C until analysis. The levels of ACTH and CORT were measured by using an ELISA test kit (Elabscience, Houston, TX, USA, Cat. No. E-EL-0160 and Cat. No. E-EL-R0048, respectively) according to the manufacturer’s instructions. Results were expressed as pg/mL and duplicate measurements were performed, and the mean value was calculated for each sample.

#### 4.3.2. Measurement of Melatonin Receptors in the Hippocampus

After decapitation, the two isolated hippocampi were placed in liquid nitrogen for snap freezing and further analysis. Following the described homogenization and centrifugation procedures, melatonin receptor levels were determined using an ELISA kit (Abbexa, Cambridge, UK, Cat. No. 533982 for 1A receptors and Cat. No. 533985 for 1b receptors). Melatonin receptor levels were expressed as ng/mg protein.

### 4.4. Statistical Analysis

Data are presented as mean ± SEM. After checking the assumptions of normality of data distribution and homogeneity of variance, differences between groups were analyzed by parametric (one-way ANOVA) or nonparametric (Kruskal–Wallis on ranks) tests with factor group (3 levels: LD-veh, LL-veh, and LL-mel) followed by either Dunn–Sidak or the Mann–Whitney U post hoc test when significant effect was detected. Two-way ANOVA test was used for analysis of melatonin levels in plasma with factors phase (2 levels: light and dark) and group. Statistically significant differences were accepted at *p* ≤ 0.05.

## 5. Conclusions

Prenatal CLE causes disturbed behavior with anxiety- and depressive-like responses in both adult male and female offspring. Impaired emotional status of the offspring was not related to changes in the melatonin system but increased plasma CORT that correlated with the missing sex dimorphism. The beneficial impact of prenatal melatonin treatment on impaired behavior of the offspring with prenatal CLE was sex-specific and stronger in male than female offspring suggesting the crucial role of up-regulated melatonin MT_1A_ and MT_1b_ receptors in the hippocampus that may be involved in the regulation of anxiety- and depressive-like responses.

## Figures and Tables

**Figure 1 ijms-26-01036-f001:**
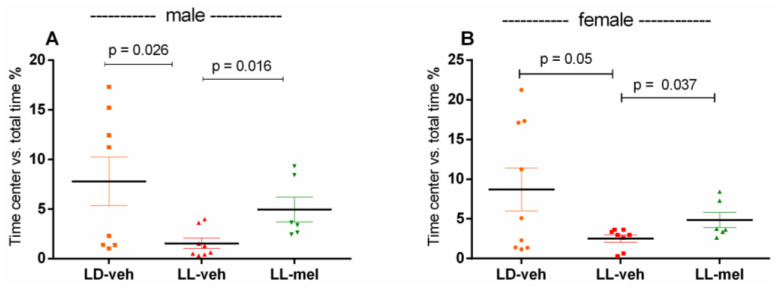
Time spent in the central zone vs. total time in percentage of male (n = 6–8) (**A**) and female (n = 6–9) (**B**) offspring exposed to prenatal light/dark (LD) and light/light (LL) regime, and treatment with vehicle (veh) and melatonin (mel) in the open field test. Data are presented as mean ± S.E.M. One-way ANOVA shows a main group effect: male: [F(2,23) = 3.734, *p* = 0.1316] (**A**); female: [F(2,23) = 8.875, *p* = 0.0017] (**B**).

**Figure 2 ijms-26-01036-f002:**
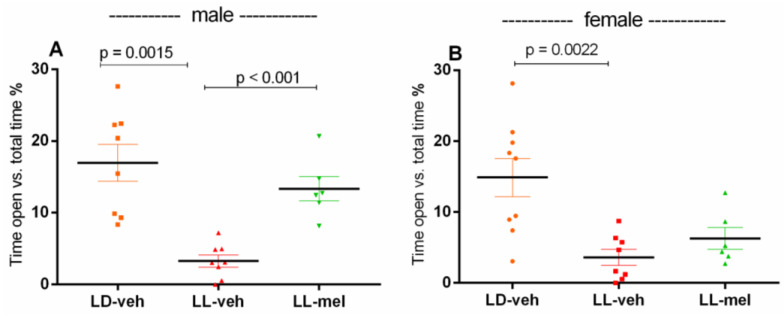
Time spent in the open arms vs. total time of male (**A**) and female (**B**) offspring exposed to prenatal light/dark (LD) and light/light (LL) regime, and treatment with vehicle (veh) and melatonin (mel) in the elevated plus maze test (n = 8). One-way ANOVA showed a main group effect: male: [F(2,23) = 9.611, *p* < 0.01]; female: [F(2,23) = 8.875, *p* = 0.006].

**Figure 3 ijms-26-01036-f003:**
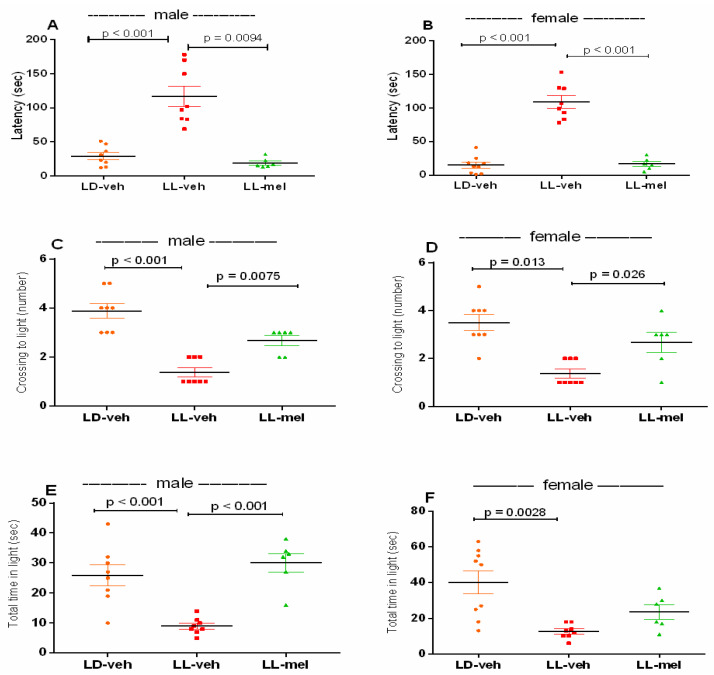
Latency (sec) to move from dark to light compartment, number of crossings to light compartments, and total time spent in light compartments of male (**A**,**C**,**E**) and female (**B**,**D**,**F**) offspring exposed to prenatal light/dark (LD) and light/light (LL) regimens, and treatment with vehicle (veh) and melatonin (mel) in the light/dark test. Data are presented as mean ± S.E.M. One-way ANOVA showed a main group effect: latency—male: [F(2,23) = 8.082, *p* = 0.0028] (**A**); female: [F(2,23) = 21.42, *p* < 0.001] (**B**); crossing to light—male: [F(2,23) = 8.723, *p* = 0.0026] (**C**); female: [F(2,23) = 6.688, *p* = 0.0045] (**D**); total time in light—male: [F(2,23) = 12.90, *p* < 0.001] (**E**); female [F(2,23) = 7.603, *p* = 0.0035] (**F**).

**Figure 4 ijms-26-01036-f004:**
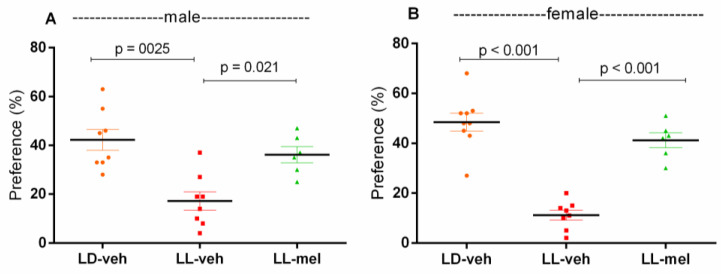
Depressive-like behavior as assessed by preference for sucrose solution in male (**A**) and female (**B**) offspring exposed to prenatal LD and LL regimens, and treatment with veh and mel as measured in the sucrose preference test. Data are presented as mean ± S.E.M. One-way ANOVA showed a main group effect: male: [F(2,23) = 7.429, *p* = 0.0031] (**A**); female: [F(2,23) = 43.12, *p* < 0.001] (**B**).

**Figure 5 ijms-26-01036-f005:**
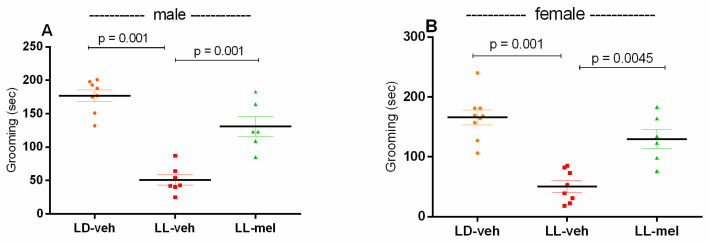
Depressive-like behavior as assessed by grooming duration in male (**A**) and female (**B**) offspring exposed to prenatal LD and LL regimens, and treatment with veh and mel as measured by the splash test. Data are presented as mean ± S.E.M. One-way ANOVA showed a main group effect: male: [F(2,23) = 20.08, *p* < 0.001] (**A**); female: [F(2,23) = 16.64, *p* < 0.001] (**B**).

**Figure 6 ijms-26-01036-f006:**
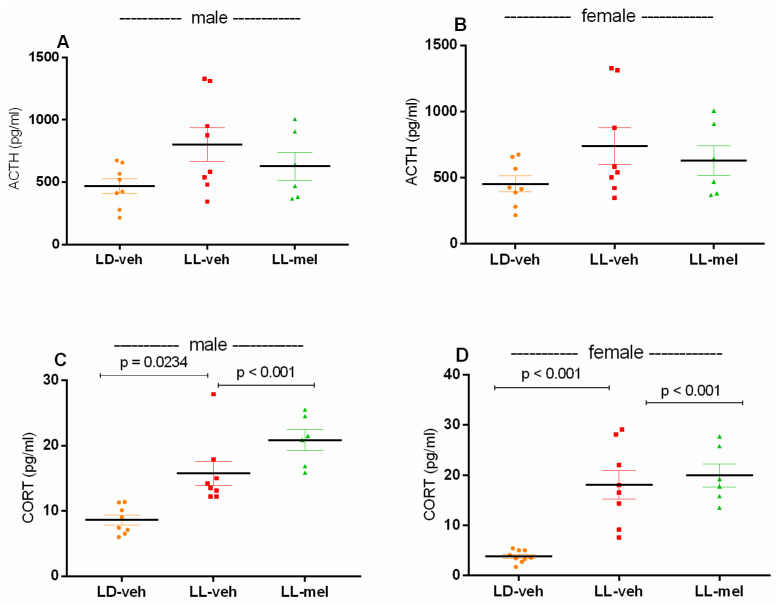
Adrenocorticotropic hormone (ACTH) and corticosterone (CORT) levels in male and female offspring exposed to prenatal LD and LL regimens, and treatment with veh and mel measured by ELISA. Data are presented as mean ± S.E.M. One-way ANOVA showed a main group effect: male: [F(2,23) = 10.92, *p* = 0.023] (**C**,**D**); female: [F(2,23) = 19.00, *p* = 0.023] (**A**,**B**).

**Figure 7 ijms-26-01036-f007:**
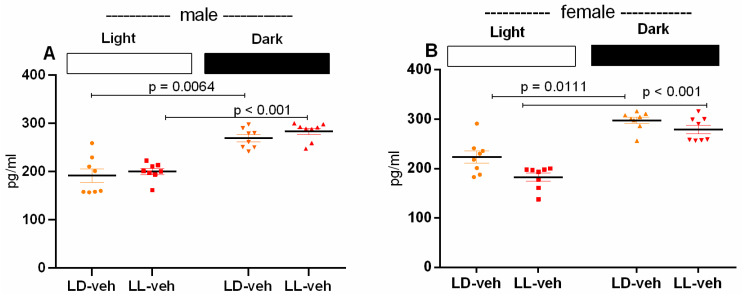
The expression of plasma melatonin levels of male (**A**) and female (**B**) offspring exposed to prenatal LD and LL regime, and treatment with veh and mel measured by ELISA. Light and dark periods are represented by open and black rectangles, respectively. Data are presented as mean ± S.E.M. Two-way ANOVA showed a main phase effect: [F(1,63) = 74.432, *p* < 0.001] (**A**,**B**).

**Figure 8 ijms-26-01036-f008:**
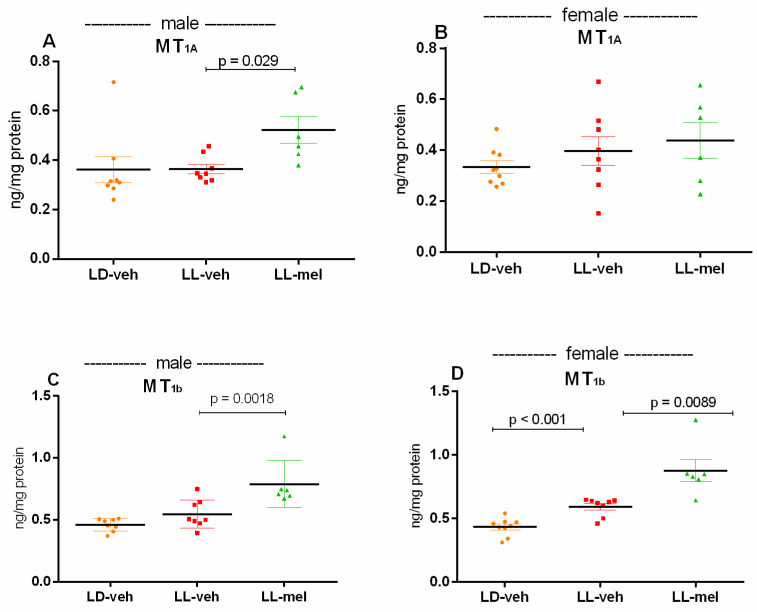
Expression of melatonin receptors MT_1A_ (**A**) and MT_1b_ (**B**) in the hippocampus of male and female offspring of females exposed to prenatal LD and LL regimens, and treatment with veh and mel measured by ELISA. Data are presented as mean ± S.E.M. One-way ANOVA showed a main group effect: male: [F(2,23) = 2.508; *p* = 0.1149] (**A**); [F(2,23) = 9.638; *p* = 0.0020] (**C**); female: [F(2,23) = 15.43; *p* = 0.0002] (**B**); [F(2,23) = 23.93; *p* < 0.001] (**D**).

## Data Availability

The data that support the findings of this study are available from the corresponding author upon reasonable request.

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
