# Peer review of "Prenatal Constant Light Exposure Induces Behavioral Deficits in Male and Female Rat Offspring: Effects of Prenatal Melatonin Treatment"

_ijms, 2025, doi:10.3390/ijms26031036_

Round 1
Reviewer 1 Report
Comments and Suggestions for Authors
The topic is highly engaging and addresses an area that can significantly enhance awareness of overlooked critical aspects, which tend to diminish the desired outcomes. In particular, it emphasizes maternal stress during pregnancy and its effects on offspring. While the study is well-structured overall, greater focus on specific missing details would further strengthen its impact.
The manuscript requires further work before it is ready for publication, including addressing specific line comments:
Title: The title is too long and the main result is unclear.
Keywords: The study discusses sexual dimorphism extensively; it may be appropriate to add this term to the keywords.
Line 92: The reference is inserted inconsistently compared to the others [n°].
Results:
- In some graphs, the standard deviation is not clearly represented (e.g., Fig. 1A, Fig. 2A).
- In some graphs, the significance markers could be made more visible.
- All captions below the figures is quite heavy and repetitive, as it overlaps with the text in the results section. I suggest adding bars to show significance between the histograms and including the p-value there.
- Some graphs have clear titles, but others, such as Fig. 5, 6, 7, and 8, are less impactful. I recommend adding titles to these graphs for clarity.
- In Figure 7, I suggest adding labels to the bars indicating light and dark conditions.
Discussion:
Lines 230- 238: the text is repetitive compared to the content in the introduction. I recommend summarizing this information.
Line 264: Since you have already mentioned the important limitation of the study, I would suggest simplifying the sentence by using "our data suggest that" instead of "we found."
Materials and Methods:
Line 298: What is the number of animals used?
Lines 345-358: The explanation of the ELISA test method is not very informative for readers who are unfamiliar with the type of kit and the processing steps. I recommend adding the following important details:
- Were the tests performed on a specific part of the blood, such as plasma or serum? How was the blood processed?
- What is the sensitivity of the kits used?
Line 360: Were the samples tested in duplicate or triplicate?
Questions: Have you considered performing a statistical comparison between sexes?
In the Figure 7 caption, you mention a three-way ANOVA, but only a two-way ANOVA is described in the text. Please correct this.
References
I suggest checking the reference numbers, as one citation does not have a number in the text
Author Response
Thank you for the careful evaluation of our manuscript. We have revised the manuscript taking into account the suggested modifications. All changes in the MS are highlighted by track changes.
Reviewer #1
Point #1 The topic is highly engaging and addresses an area that can significantly enhance awareness of overlooked critical aspects, which tend to diminish the desired outcomes. In particular, it emphasizes maternal stress during pregnancy and its effects on offspring. While the study is well-structured overall, focusing more on specific missing details would further strengthen its impact. The manuscript requires further work before it is ready for publication, including addressing specific line comments. Title: The title is too long, and the main result is unclear.
Response: We fully agree with this relevant remark. The title of the mns was edited.
Point #2: Keywords: The study discusses sexual dimorphism extensively; it may be appropriate to add this term to the keywords.
Response: We are thankful for this suggestion. The term “sexual dimorphism” has been added as a keyword.
Point #4: Line 92: The reference is inserted inconsistently compared to the others [n°].
Response: The mentioned reference was corrected.
Point #5: Results: In some graphs, the standard deviation is not clearly represented (e.g., Fig. 1A, Fig. 2A). In some graphs, the significance markers could be made more visible.
Response: We are grateful for all the reviewer's comments regarding the figures. They have been modified to have the same size and thickness (SEM). In addition, the significance markers have been modified for a more straightforward perception.
Point #6: All captions below the figures are pretty heavy and repetitive, as they overlap with the text in the results section. I suggest adding bars to showthe significance between the histograms and including the p-value there.
Response: We fully agree with the note of the Reviewer. Following this recommendation, we simplified the text to figures and inserted p-values above the scatter plots in the figures.
Point #7: Some graphs have clear titles, but others, such as Fig. 5, 6, 7, and 8, are less impactful. I recommend adding titles to these graphs for clarity.
Response: One of the Reviewers recommended putting “male” and “female” above the scatter plots as titles and the name of the test (previously used by us as a title) to be replaced in the chapter of the figure.
Point #8: In Figure 7, I suggest adding labels to the bars indicating light and dark conditions.
Response: We are thankful for this suggestion. For clarity, this text was added in the chapter on the figure (Light and dark periods are represented by open and black rectangles, respectively).
Point #9: Discussion: Lines 230- 238: the text is repetitive compared to the content in the introduction. I recommend summarizing this information.
Response: The mentioned sentence was removed from the Discussion (line 246-248).
Point #10: Line 264: Since you have already mentioned the critical limitation of the study, I would suggest simplifying the sentence by using "our data suggest that" instead of "we found."
Response: Following the recommendation of the Reviewer the mentioned sentence was corrected. We replaced "we found" by "our data suggest that".
Point #11: Materials and Methods: Line 298: What number of animals are used?
Response: The number of animals per group was indicated in Methods and the chapters of all the figures.
Point #12: Lines 345-358: The explanation of the ELISA test method is not very informative for readers unfamiliar with the type of kit and the processing steps. I recommend adding the following important details:
Response: The description of ELISA test was modified for better clarity.
Point #13: Were the tests performed on a specific part of the blood, such as plasma or serum? How was the blood processed?
Response: The blood was taken from the tail vein. The procedure for extraction of plasma by centrifugation was added in the text.
Point #14: What is the sensitivity of the kits used?
Response: This information was inserted in text in Methods.
Point #15: Line 360: Were the samples tested duplicates or triplicates?
Response: The samples were tested in duplicate as indicated in the text in Methods (line 361).
Point #16: Questions: Have you considered performing a statistical comparison between sexes?
Response: We decided to evaluate the effects of melatonin separately in male and female offspring for easier interpretation, and data described in the figures for male and female offspring were given separately.
Point #17: In the Figure 7 caption, you mention a three-way ANOVA, but only a two-way ANOVA is described in the text. Please correct this.
Response: We are thankful for this remark. This technical error was corrected in the text in Fig 7.
Point #18: References I suggest checking the reference numbers, as one citation does not have a number in the text
Response: All references were checked again, and a number was given for the mentioned citation.

Reviewer 2 Report
Comments and Suggestions for Authors
This study provides valuable insights into the impact of prenatal constant light exposure (CLE) on behavioral and physiological outcomes in adult rat offspring and highlights the potential of melatonin as a therapeutic intervention. The use of well-designed behavioral tests and biochemical analyses to assess anxiety, depression-like behaviors, and melatonin system regulation is commendable. I have some concerns:
Major: The authors propose that the beneficial effects of prenatal melatonin treatment on behavioral impairments induced by prenatal CLE exhibit sex-specific differences, with male offspring showing a stronger response than females. They suggest that the upregulation of melatonin MT1A and MT1B receptors in the hippocampus may play a key role in modulating anxiety- and depression-like behaviors. It would be more convincing if the authors could selectively downregulate MT1A and MT1B receptors in the hippocampus and then assess whether the effects of melatonin are abolished.
Minor:
—I find the title overly lengthy and difficult to comprehend. I suggest further condensing it for clarity.
—It is recommended to include scatter plots for individual data points alongside the bar charts.
—The statistical methods section does not specify how data distribution was assessed. Were the data tested for normality? Are they suitable for parametric analysis?
—The font size of the y-axis titles in different panels of Figure 3 is inconsistent. Similar issues are present in Figures 6 and 8.
—There are inconsistencies in figure citation formatting. For instance, Line 160 cites "Figure 4," whereas Line 162 cites "Figure 5A, B."
—In the abstract: under CLE conditions, melatonin levels in male and female offspring remain unchanged. It would be helpful to simultaneously emphasize that their rhythmicity is maintained.
Author Response
Reviewer #2
Point #1: Comments and Suggestions for Authors
Major: The authors propose that the beneficial effects of prenatal melatonin treatment on behavioral impairments induced by prenatal CLE exhibit sex-specific differences, with male offspring showing a stronger response than females. They suggest that the upregulation of melatonin MT1A and MT1B receptors in the hippocampus may play a key role in modulating anxiety- and depression-like behaviors. It would be more convincing if the authors could selectively downregulate MT1A and MT1B receptors in the hippocampus and then assess whether the effects of melatonin are abolished.
Response: We are grateful for the reviewer's suggestion. The results of the present study, including the finding that the beneficial effects of prenatal melatonin may be related to the upregulation of MT receptors, give us the background to continue studies in this model, including using an approach to downregulate MT receptors in the hippocampus.
Point #2: Minor:
—I find the title overly lengthy and difficult to comprehend. I suggest further condensing it for clarity.
Response: We fully agree with this relevant remark. The title of the mns was edited.
Point #3: —It is recommended to include scatter plots for individual data points alongside the bar charts.
Response: Following this reviewer's remark, we clarified the figures by replacing bars with scatter plots.
Point #4: —The statistical methods section does not specify how data distribution was assessed. Were the data tested for normality? Are they suitable for parametric analysis?
Response: We agree with this remark of the Reviewer and indicated if data were assessed by parametric or non-parametric test (Kruskal-Wallis). The text was edited following this note from the Reviewer in Method/Statistical analysis.
Point #5: —The font size of the y-axis titles in different panels of Figure 3 is inconsistent. Similar issues are present in Figures 6 and 8.
Response: All figures were corrected in a way to have the same size of y-axis title.
Point #6: —There are inconsistencies in figure citation formatting. For instance, Line 160 cites "Figure 4," whereas Line 162 cites "Figure 5A, B."
Response: The mentioned citations of Fig. 4 and Fig. 5A,B respond the mentioned figures in the text and they give information for depressive-like responses.
Point #7: —In the abstract: under CLE conditions, melatonin levels in male and female offspring remain unchanged. It would be helpful to emphasize that their rhythmicity is maintained simultaneously.
Response: We are thankful for this advice. This information was also included in the new version of the abstract.

Reviewer 3 Report
Comments and Suggestions for Authors
Review IJMS-3416908
The manuscript by Stoyanova et al describe their study on the effect of prenatal constant light exposure (CLE) and melatonin treatment on anxiety- and depression-like behaviors and melatonin system in male and female adult rat offspring. They identified some significant detrimental effect of CLE, some of which were elevated by melatonin treatment. They have also determined plasma adrenocorticotropic hormone (ACTH), corticosterone (CORT) and melatonin expression as well as hippocampal MT1A and MT1b receptor expression.
Overall, I don’t have any major concerns with the experimental design, and I think the study provides an important contribution to the field and will be interesting read for the IJMS readership.
Minor comments
ALL the data generated in this study, behavioral and biochemical, should be made available for the readers via public depositories such as Zenodo or Dryad.
line 39: references missing
line 42-50: the section does not read as part of the introduction, and looks as if it has been copied from a another essay. e.g " the evidence... is also reviewed" what does it mean "finally, a multilevel .. approach .... is presented"
line 47: the Introduction should start from line 47
line 70: replace "rodents" with "mammals
line 92: omit full stop after 'offspring'
line 104: report the statistical test, the statistics and degree of freedom.
line 109, Figure 1. the color is not necessary here, neither the different pattern for males and females. Move the 'male' 'female' headers up into the plot space. move the main header 'open field test' to the figure caption (first sentence).
line 112-114 keep the report of F and p-value together for each test.
line 116: replace 'with a history" with 'exposed to’
figure 2. See comment on previous figure. move the main header to the caption. sort the statistics reporting and remove the color
line 133: text does not make sense 'showed increased 1) latency
line 142, figure 3, see the comments on previous figure
line 186: the result was not statistically significant, so 'showed a tendency' is meaningless. Please reword the sentence.
line 208: the 3-way anova should be explained. what are the factors? any of the interactions was tested? the circle symbols for significance are confusing. Please use the same symbols as in previous figures.
line 223: the reader does not know what is the 'Group' factor.
line 241. "no sex dimorphism.....in the two sexes". the sentence seems to be grammatically wrong. please amend.
line 289. This sentence is too long. Please rewrite it.
line 302: Describe the light source, and light intensity.
line 354: replace 'the isolated two' with 'the two isolated'
line 362: In the result section (in one of the figure captions) a three-way ANOVA is mentioned, but this is not explained here. also, please explain what 'phase' and 'group' signify.
line 365: the word 'related' seems not right here. please reword.
Comments on the Quality of English LanguageThe text will benefit from professional editing. In my comments, I pointed a few issues to the authors.
Author Response
Review # 3
Point #1: ALL the data generated in this study, behavioral and biochemical, should be made available for the readers via public depositories such as Zenodo or Dryad.
Response: Following the advice of the Reviewer, we will take action to submit data from this study to the public depository.
Point #2: line 39: references missing
Response: Corrected.
Point #3: line 42-50: the section does not read as part of the introduction, and looks as if it has been copied from a another essay. e.g " the evidence... is also reviewed" what does it mean "finally, a multilevel .. approach .... is presented". line 47: the Introduction should start from line 47
Response: We fully agree with this comment. The first paragraph of the Introduction was edited following the relevant notes of the Reviewer
Point #4: line 70: replace "rodents" with "mammals
Response: Corrected.
Point #5: line 92: omit full stop after 'offspring'
Response: Corrected.
Point #6: line 104: report the statistical test, the statistics and degree of freedom.
Response: Full description of statistical data (F =…, p =…) were shown in the text to figures, while in the Result text, only p values were given. We prefer this style of presentation of experimental data for more straightforward perception of results and understanding of their meaning.
Point #7: line 109, Figure 1. the color is not necessary here, neither the different pattern for males and females. Move the 'male' 'female' headers up into the plot space. move the main header 'open field test' to the figure caption (first sentence).
Response: Following the reviewers' advice, we have completely redesigned all figures. To make the data presentation more transparent in the new version, we used scatter plots instead of bars for the figures. We also moved the "male" and "female" headings into the plot space.
Point #8: line 112-114 keep the report of F and p-value together for each test.
Response: The Reviewer #1 recommended to insert “bars to show significance between the histograms and including the p-value there” because “the captions below the figures is quite heavy and repetitive, as it overlaps with the text in the results section”. We decided to change the chapter on all the figures and add the p-value to the top of the scatter plots.
Point #9: line 116: replace 'with a history" with 'exposed to’
Response: Corrected. We also did this to replace the text with figures.
Point #10: figure 2. See comment on previous figure. move the main header to the caption. sort the statistics reporting and remove the color
Response: Corrected.
Point #12: line 133: text does not make sense 'showed increased 1) latency
Response: The word mentioned above was replaced by “had”.
Point #13: line 142, figure 3, see the comments on previous figure
Response: Corrected. All the figures and their chapters were edited according to the reviewers' notes.
Point #14: line 186: the result was not statistically significant, so 'showed a tendency' is meaningless. Please reword the sentence.
Response: The sentence was edited.
Point #15: line 208: the 3-way anova should be explained. what are the factors? any of the interactions was tested? the circle symbols for significance are confusing. Please use the same symbols as in previous figures.
Response: For the results described in Fig. 7, we used two-way ANOVA, and there was a technical error in the first version when writing three-way ANOVA. The factors were Phase and Group. Only the factors Phase showed a main effect, and F = result was reported in the chapter to figure. The factors used were also mentioned in the Methods/Statistical analysis.
Point #16: line 223: the reader does not know what is the 'Group' factor.
Response: The three levels of the factor Group were reported in the Method section/ Statistical analysis.
Point #17: line 241. "no sex dimorphism.....in the two sexes". the sentence seems to be grammatically wrong. please amend.
Response: Corrected.
Point #18: line 289. This sentence is too long. Please rewrite it.
Response: We have edited this text. We have split it into two separate sentences.
Point #19: line 302: Describe the light source, and light intensity.
Response: The light source and light regime were described in the new version of the manuscript in Method section 4.1. The Animals and experimental design.
Point #20: line 354: replace 'the isolated two' with 'the two isolated'
Response: This technical error was removed.
Point #21: line 362: In the result section (in one of the figure captions) a three-way ANOVA is mentioned, but this is not explained here. also, please explain what 'phase' and 'group' signify.
Response: The factors and their levels were described in Statistical analysis in Method section.
Point #22: line 365: the word 'related' seems not right here. please reword.
Response: Corrected.
Point #23: Comments on the Quality of English Language
The text will benefit from professional editing. In my comments, I pointed a few issues to the authors.
Response: The manuscript has been carefully checked for errors in style and grammar, and the necessary corrections have been made.

Round 2
Reviewer 1 Report
Comments and Suggestions for Authors
I have no further comments.
Reviewer 2 Report
Comments and Suggestions for Authors
This article indeed provides some valuable insights. The authors have carefully addressed some of the questions I raised. I recommend it for acceptance and publication.